# China's plastic import ban increases prospects of environmental impact mitigation of plastic waste trade flow worldwide

Zongguo Wen [1✉], Yiling Xie [1], Muhan Chen[1] & Christian Doh Dinga[1]

Since the late 1990s, the trend of plastic waste shipment from developed to developing countries has been increasing. In 2017, China announced an unprecedented ban on its import of most plastic waste, resulting in a sharp decline in global plastic waste trade flow and changes in the treatment structure of countries, whose impacts on global environmental sustainability are enormous but yet unexamined. Here, through the life cycle assessment (LCA) method, we quantified the environmental impacts of changes in the flow patterns and treatment methods of 6 types of plastic waste in 18 countries subsequent to the ban. In the short term, the ban significantly improved four midpoint indicators of environmental impact, albeit contributed to global warming. An annual saving of about 2.35 billion euros of eco-cost was realized, which is equivalent to 56% of plastic waste global trade value in 2017. To achieve global environmental sustainability in the long run, countries should gradually realize the transition from export to domestic management, and from landfill to recycling, which would realize eco-costs savings of about 1.54–3.20 billion euros.

[1] State Key Joint Laboratory of Environment Simulation and Pollution Control (SKLESPC), School of Environment, Tsinghua University, Beijing 100084, China.
✉email: wenzg@tsinghua.edu.cn

Over the last few decades, plastics have been vastly used as an important material with an alarmingly growing rate of production globally[1]. The cumulative global production of plastics has been approximately 8.3 billion tons since 1950, half of which was produced in the past 13 years (more than 300 million tons annually in recent years)[2–4]. However, the recycling rate of plastic varies widely between countries and is still about 9% globally. Most plastic ends up in landfills, incineration plants, or is mismanaged[2,5].

There are two main ways to deal with plastic waste pollution: domestic management and export. Due to the environmental risks and economic benefits, global plastic waste trade flows from high-income countries to low-income countries have become a routine since the late 1990s[6,7]. Although most of plastic is recyclable and to some extent, can potentially mitigate the shortage of resources in developing countries, it has however caused significant environmental impacts from the trade and disposal processes[8,9], not to mention that plastic waste can even contain hazardous substances[10]. The Basel Convention enacted in the 1980s was adopted in a bid to guarantee the environmental security of developing countries vulnerable to waste pollution from developed countries. However, the Basel Convention has not strongly affected the international plastic trade market as it should be/as we wish[10,11], because it didn't offer a clear definition of hazardous waste and some major waste exporters refused to sign on to it. The treatment system and technical advancement of plastic waste management differ between countries, that is, waste disposal in developing countries is often at a higher environmental cost[6] and it is hard to intuitively judge whether such waste trade flows are globally sustainable.

China was the main importing country of plastic waste[6] and the largest plastic producer in the world[4]. Before the ban, Chinese annual imports of plastic waste reached 8.88 million tons, with as much as 70.6% buried or even mismanaged, triggering a series of environmental problems[8,12]. To mitigate this situation, on July 27, 2017, China issued a new ban named Prohibition of Foreign Garbage Imports: the Reform Plan on Solid Waste Import Management (herein referred to as "the China ban"), banning its import of 24 types of solid waste which included plastic waste. This abrupt ban prompted changes in both the short and long run in global plastic waste trade flow patterns as well as plastic waste treatment systems and mechanisms in many countries. From a global sustainability perspective, the resulting environmental impacts do not only require serious attention but also need to be quantitatively assessed.

Current researches on the impacts of the China ban are either conducting qualitative discussions[6,13], focusing solely on its trade effects[14–16], or quantifying the value-added loss of China and increased requirement of waste treatment capacity for other economics through a hypothetical extraction method[17]. However, a handful of these studies took the technical differences among countries into account from a global sustainability perspective and provided specific suggestions on the trend of the international plastic waste trade. Because plastic waste treatment methods and technologies vary between countries, likewise the environmental impacts of treating 1 kg of plastic waste have great disparity. The incineration and recycling rates in developed countries are generally much higher than those in developing countries with landfills as their main disposal means, which implies that the changes in global plastic waste trade flow influence the already existing environmental impacts to a large extent. In addition, the environmental impacts of treatments of plastic wastes differ from type to type, and that's why the environmental impact assessment needs to be further refined.

On the basis of the research gap mentioned above, this study first identified the de facto changes in global plastic waste trade flow patterns in the aftermath of the ban (From Baseline Scenario to 2018 Scenario), mainly based on data from The UN Commodity Trade Statistics Database (referred to as "Comtrade"[18]). Sample countries included China, "17 countries (and regions)" which accounted for 92% of plastic waste imported to China, and "other countries". After quantifying the change in pattern of global plastic waste trade flow, we further evaluated the de facto environmental impacts and the corresponding Eco-cost of these changes in global plastic waste trade flow patterns through life cycle assessment (LCA), in which six types of plastics were considered. In this step, we fully took into account changes in the environmental impact due to technical differences among countries, including diversities of technological structure and advancement. Last but not the least, on basis of the "Forced Mechanism" of the ban, it is predicted that the global plastic waste trade will further plummet while the waste treatment capacities, especially the recycling treatment, of countries will increase. To fill the gap of quantitative assumption and explore the long-term environmental ramifications, we adopted Scenario Analysis (SA) to develop two types of prediction scenarios and integrate them: (1) Export Reduction Scenarios assume that the exports of plastic waste of developed countries and developing countries plunged by 50% simultaneously or separately against 2018 volume, and (2) Recycling Rate Promotion Scenarios premise that the recycling rates of all countries are increased by 20%, 50%, and 100% in comparison to the 2018 scenario.

Through our work, we displayed the global trade flows of six types of plastic wastes before and subsequent to the China ban, calculated the environmental impacts and the eco-costs of five relevant midpoint indicators of various scenarios emanating from the China ban, and presented the most recommended scenario. Targeted policy implications are put forward based on results.

## Results

**Changes in plastic waste trade flow pattern subsequent to the China ban.** Before the ban, mainland China was undoubtfully the single largest importer of plastic waste, importing about 55.7% of world's plastic waste (14,304,561 tons for the world). Hong Kong, China, was one of the most prominent transshipment ports for plastic waste destined for mainland China, transferring about 3,184,176 tons of plastic waste per annuum (22% of world's trades) to China. The USA, Japan and seven European countries (herein referred to as "Europe 7"), respectively exported 77.9%, 87.6%, and 57.5% of plastic waste to China (including Hong Kong). In addition, five Southeast Asian countries (herein referred to as "Southeast Asia 5") were also destinations for plastic waste. Figure 1a presents the Sankey diagram of the global trade flows of six types of plastic waste prior to the ban (Baseline Scenario). Polyethylene (PE) accounted for 37% of the plastic waste trade flow among the 18 sample countries (11,404,697 tons in total) and ranked first. Polypropylene (PP), polyethylene terephthalate (PET), and others accounted for 23%, 12%, and 14% respectively, while polystyrene (PS) and polyvinyl chloride (PVC) were under 8%. The market share of PE in Hong Kong was as high as 46.2%, which made the PE flow from Hong Kong to China particularly prominent in Fig. 1a.

The world's total plastic waste trade flow plunged by 45.5% in 2018 compared to the Baseline Scenario. Global plastic waste trade flow following the ban (2018 Scenario) is depicted in Fig. 1b, the flow of which is well-proportioned. Comparing Fig. 1b to Fig. 1a, Hong Kong's dominance of exports dwindled. The disparities in global trade patterns of plastic waste in the two scenarios reflect the complex changes in trade flow between countries. The exports of four countries, namely Japan, the USA, Germany, and the UK, accounted for 46.1% of the world trade

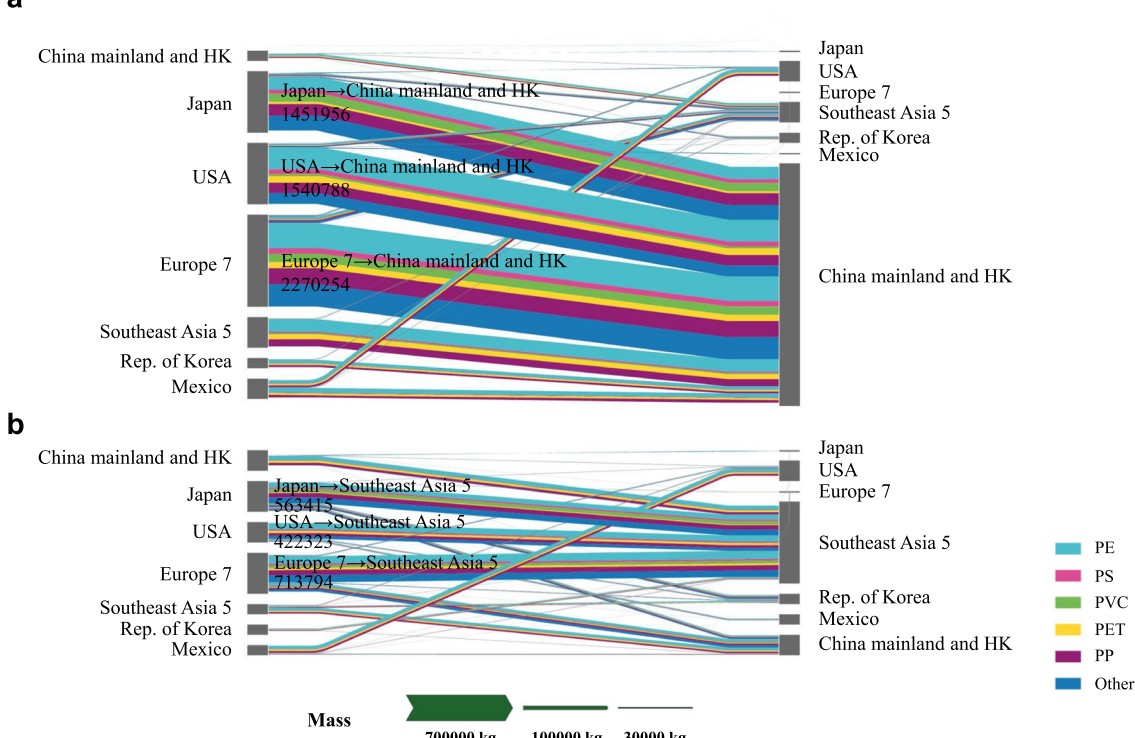

**Fig. 1 The trade flows of six types of plastic waste under two scenarios. a** presents the global trade flows of six types of plastic waste prior to the ban (Baseline Scenario), while **b** presents the flows subsequent to the ban (2018 Scenario). The unit of the flows is tons. The data source of **a** and **b** is shown in Supplementary Tables 1 and 2. The flows of six types in the Figure are based on estimation (Supplementary Table 4).

flow of plastic waste (7,800,583 tons for the world). Malaysia, instead of China, became the top export destination for most countries and had the largest import flow of plastic waste (12.2% of the total) among these 18 sample countries. PE remained the largest part of the plastic waste trade flow among the 18 sample countries, but its share plummeted to 33%, while the change rates of flows of PVC, PET, and others worldwide are +23.8%, −20.1%, and +41.8% respectively, mainly resulting from the slump in exports of Hong Kong and Southeast Asia 5. As depicted in Supplementary Fig. 1, after calculating the $\lg(TF^{2018}/TF^{\text{baseline}})$ of sample countries, we ascertained that, in 18 countries, only China and Belgium had increments in exports.

The sharp decrease in import flow of China (to 4.6% of Baseline) indicates that the ban is remarkably stringent and fulfills its primary aim of reducing plastic waste imports. China's exports in 2018 were about twice that of Baseline, which is consistent with Qu et al. (2019)'s opinion that China may grow to be an exporter of solid waste itself in the future[6]. As the transshipment port of mainland China, Hong Kong experienced a significant decrease in both imports (−88.7%) and exports (−91.3%). Japan, the USA, and Europe 7 had similar changes in their exports following the ban, specifically, their exports were down by 39.2%, 54.1%, and 29.9%, all due to their decrease in exports to China and Hong Kong. Because developed countries with large export flow in the Baseline Scenario could not fully get rid of their produced plastic waste internally in a short period of time, and their domestic recycling markets had not yet been fully constructed, they consequently chose to stock up or transfer plastic waste to third countries other than China. Their major export destinations were swiftly switched to Southeast Asia, which appeared in our results as that proportion of exports to Southeast Asia 5 soared from 4.34% to 55.9% for Japan, from 5.24% to 46.5% for the USA, and from 6.1% to 33.0% for Europe 7. The import flow of Europe 7 had an increase on account of the growth of the trade flow among

European countries (Supplementary Fig. 2 and Supplementary Fig. 3). Similar to Japan, the USA, and Europe 7, the exports of The Republic of Korea and Mexico to Southeast Asia 5 increased significantly. However, these two countries both roughly doubled their imports, which were mainly from the USA or Japan.

The total exports of Southeast Asia 5 decreased to 49.9% in 2018. Although globally, the unit price of plastic waste did not change significantly in 2018 compared to 2017. Due to the lack of comprehensive import restrictions and a certain degree of domestic resource demand, the import flows of Southeast Asia 5 roused to 3.62 times that of the Baseline Scenario, which mainly came from Japan (25.8%), the USA (19.4%), Germany (11.7%), Hong Kong (10.3%), and the UK (9.8%).

Based on the export data available in Comtrade as of June 2020, the sum of plastic waste trade flows from all countries that have reported export data for these two consecutive years to Southeast Asia 5 in 2019 decreased by 32% compared to 2018 Scenario (1948554 t for 2018, and 1331851 t for 2019.) This result is consistent with the 8 sample countries that have reported export data of 18 sample countries to Southeast Asia 5 of 2019 (Source data is shown in Supplementary Table 10). It shows that there may be a downward trend for the transfer of plastic waste from developed countries to Southeast Asia after 2018. Meanwhile, the impact of the ban continued to intensify in 2019, with 8 countries' trade flows to China dropping to 41% and the export flows of the USA, UK, and Republic of Korea falling to 60%, 37%, and 46% of 2018 respectively.

To determine the relationship between the export flow changes of countries and their dependence on the Chinese market, we drew Fig. 2, according to Supplementary Table 3. The results illustrate that there is a strong negative correlation, with the R value of −0.677 ($p = 0.003$). That is to say, if a country relied on the Chinese market to a large extent, in other words, the country was exporting most of its plastic waste to China and Hong Kong,

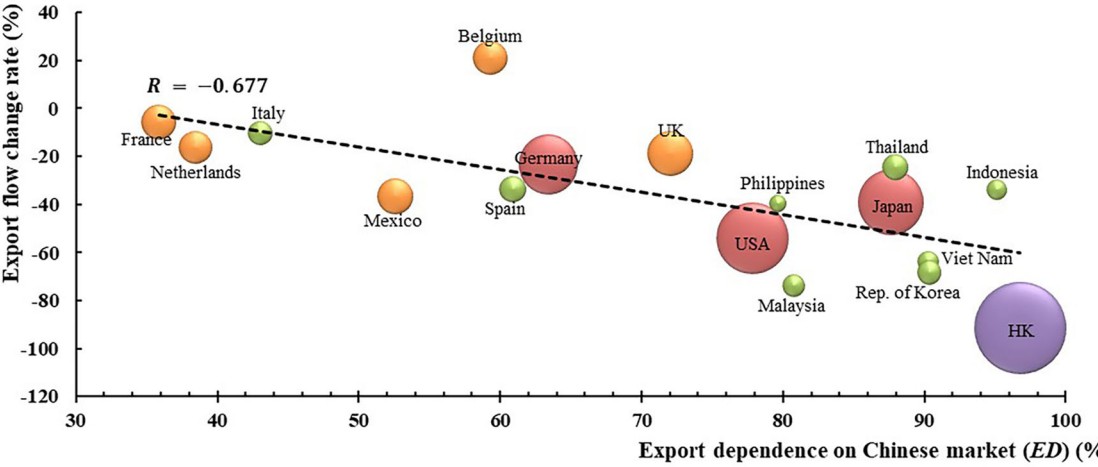

**Fig. 2 The relationship between Export flow change rate and *ED*.** Export dependence on the Chinese market (ED) is defined as the proportion of export flows to mainland China and Hong Kong in the total exports of a country under the Baseline Scenario. The bubble shape size indicates the total Baseline export flow of the countries, according to which countries are divided into four categories and identified by four colors.

its trade structure would be vulnerable and easily influenced by abrupt changes such as the China ban. This finding corresponds with Brooks et al. (2018)'s opinion that the dependence of a single importer is fragile[14].

Forced mechanism of the ban not only leads to the sharp decline in global plastic waste trade flows, but also promotes adjustments in the plastic waste treatment system structures of countries, which would influence global environmental sustainability. Therefore, life cycle assessments were further conducted after quantifying global waste trade flows.

**Environmental impacts of changes in waste trade flows and treatment system structures.** Due to the diversity in technical advancement and plastic waste treatment system structures of different countries, the unit environmental impacts of plastic waste treatment vary from country to country. We used LCA and SA to evaluate the environmental impacts and eco-cost of the China ban under different future trends and projections, taking into account domestic management, export, and transportation. Treatment of 1 kg of plastic waste was chosen as the functional unit. The system boundaries for LCA of Environmental Impact of Trade flow changes (EIT) are shown in Fig. 3. The cradle to grave end-of-life phase of plastic waste begins from the transportation of plastic waste from collection spots to disposal facilities. For domestic management and export, three common treatment options for plastic waste were considered, of which a few cases of mismanagement is approximated by sanitary landfill due to the lack of precise environmental impact data that may lead to uncertainty. Apart from these three common options, there are also some emerging techniques for plastic waste treatment, such as refuse-derived fuel production, the share of which are negligible. The EIT of one trade flow exported from Country i to Country j is defined by the formula below:

$$EIT_{ij} = (EIT_{DM})_{ij} + (EIT_{Exp})_{ij} + (EIT_{tran})_{ij}$$
$$= (EX'_{ij} - EX_{ij})(-DM_{ij} + Exp_{ij} + tran_{ij})$$

(1)

where $EX_{ij}$ is the export flow (in kg) from i to j in Baseline Scenario and $EX'_{ij}$ is that of 2018. $DM_{ij}$ and $Exp_{ij}$ are the environmental impacts of treating 1 kg of plastic waste, which is generated from Country i, by Country i and j respectively. $tran_{ij}$ is the environmental impact of transporting 1 kg of plastic waste from Country i to j, which is corresponding to the process of T1

and T2 in Fig. 3. Through literature research, the proportions of six types of plastics and three common treatment options of the sample countries are shown in Supplementary Table 4. As per the data available, there are distinctions in the shares of three treatment options under the baseline and 2018 scenarios for some countries and specifically, the landfill rate declined, recycling rate rose while incineration rate fluctuated slightly. The recycling and incineration rates of plastic waste in developed countries are generally higher than that in developing countries where landfill plays the dominant role in waste treatment. The unit impact value of treating 1 kg of plastic waste from Country i by Country j was calculated through matrix operation. After which, all the impact values in the matrix were multiplied by trade flow changes.

The environmental impact of five midpoint indicators (global warming (GW) (kg $CO_2$ eq), fine particulate matter formation (FPMF) (kg PM2.5 eq), freshwater ecotoxicity (FWE) (kg 1,4-DCB eq), human carcinogenic toxicity (HCT) (kg 1,4-DCB eq) and water consumption (WC) (m^3)) in ReCiPe resulting from the China ban for the 2018 Scenario is shown in Fig. 4. As it can be observed, the trade flow alterations contribute to GW, and more or less improved the other four indicators. $EIT_{tran}$ of all the five indicators are negative under the background of decreasing total trade flow, that is to say that the decrease in flow after the ban mitigated environmental impacts from the trade of transportation of plastic waste. However, its contribution to the impact is significant only on GW and FPMF. There are both positive and negative impact values in $EIT_{DM}$ and $EIT_{EXP}$ since the proportion and environmental impact of three treatment options of plastic waste are quite different.

The EIT of treating 1 kg of domestically produced plastic waste in each country is shown in Supplementary Fig. 4. For GW, the unit impact values of incineration are much higher than those of landfill and recycling, resulting in countries with high incineration rates having greater unit impact values. We found that there is a strong positive linear correlation between unit impact values on GW of treating plastic waste domestically and per capita GDP in 2018 of countries (R=0.772, p=0.000, data is shown in Supplementary Table 6), which implies developed countries have greater unit impact values on GW. The China ban prompted developed countries to treat more plastic waste on their own, and the resulting EIT on GW is obvious to understand. An increase in share of incineration treatment means more water consumption from electricity generation would be avoided explained by the huge savings of global flow changes in WC. As for FWE, the unit

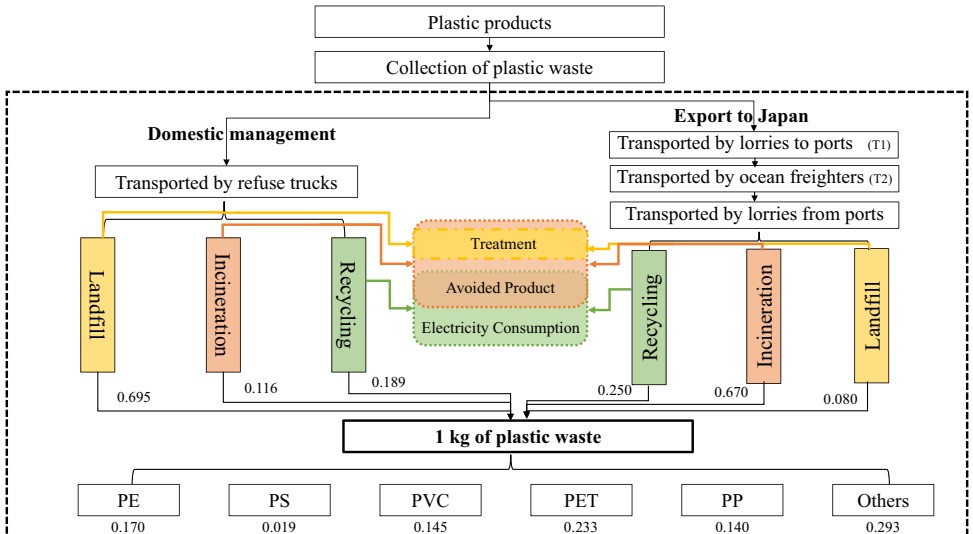

**Fig. 3 System boundaries for LCA of EIT.** Here, China's export to Japan under the Baseline Scenario is taken as an example. The dotted line is the system boundary. The proportion of different types of plastic waste generated in China is shown at the bottom of the Figure. For example, PE accounts for 0.170 kg of 1 kg of plastic waste. For every 1 kg of plastic waste (generated by China) treated domestically, the environmental impact of transportation by refuse trucks are considered together with the environmental impact from the three treatment options of China. Whereas, when exporting to Japan, both the environmental impact of sea and land transportation are considered first, then that from plastic waste treatment options in Japan.

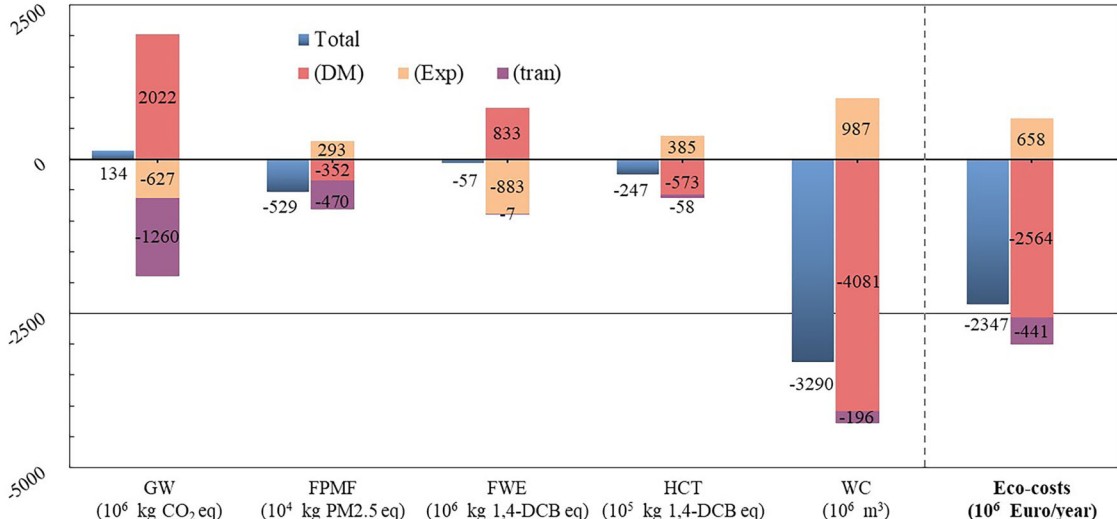

**Fig. 4 EIT and Eco-cost of the China ban for 2018 Scenario.** Note that an item has a beneficial environmental impact when its value is negative. To enhance the visibility of midpoint indicator values on the ordinate axis, the unit of each indicator was adjusted as shown in the brackets at the bottom of the Figure.

impact values of landfill and incineration are similar, with minimal differences between countries. The differences among countries for HCT is to a large extent due to the recycling rate. In addition, according to the results of the Sensitivity Analysis (Supplementary Table 7), the results of EIT for GW were mainly influenced by "unit avoided plastic production". Therefore, the optimization of recycling plastic technology is one of the underlying pillars to improve EIT for GW.

As exporters, countries with huge flow change may account for a significant share of impacts, as shown in Supplementary Fig. 5a that Japan, the USA, Hong Kong, and Germany account for the major parts of five indicators. As importers, the import changes of China, Malaysia, and Hong Kong play a leading role in total EIT, as shown in Supplementary Fig. 5b. Among them, the EIT of five indicators caused by the changes of China's import flows is in the same direction (positive and negative) as the result of the total EIT.

Due to the advanced treatment technology and structure of developed countries, the reduction of China's import flows constrained plastic waste treatment in developed countries to domestic management, which undoubtedly improves the latter four indicators. Meanwhile, the EIT of GW resulting from China's import changes is as high as 8.3 times of the total EIT. When huge quantities of plastic wastes generated in developed countries are rejected for import by China, and are subsequently transferred for domestic treatment in their respective countries of origin, a more severe GW impact will be experienced (since the incineration rate in developed countries is higher) than if they were treated in China which has a relatively higher landfilled rate. Overall, the changes of China's import flows have saved 2.51 billion euros of eco-cost. As the largest importer in 2018, the environmental impacts of the import changes in Malaysia are also significant in the total EIT.

By means of Eco-cost method, the results of five midpoints were further aggregated into a single "Eco-cost", which is the sum of the product of environmental impacts of all midpoints and its corresponding eco-cost values[19]. As shown in Fig. 4 and Supplementary Table 5, due to the global trade flow changes of plastic waste, a total of 2.35 billion (from $2.17 \times 10^9$ to $2.52 \times 10^9$) euros of eco-cost was saved annually around the world after the China ban. The eco-cost saved is equivalent to approximately 56% of the global trade value in plastic waste in 2017(4.7 billion US dollars[18]), which is calculated at the average EurUsd exchange rate in 2017. The contribution of WC accounts for the largest proportion of eco-cost savings because of the higher rates of incineration in developed countries, which avoided water consumption from electricity generation, as earlier iterated (Supplementary Fig. 6).

China has annunciated it will not only ban its imports of plastic waste, but also extend this ban to many kinds of foreign garbage in the future, which means that the effects of the China ban on global plastic waste trade flow will be relentless and persistent, with the world constrained to act appropriately to tackle these changes, hence, we referred to this situation as a "Forced Mechanism". In the near future, large amounts of waste will be disbursed to other developing countries where there is still lack of policies to restrict imports[6,8,14,16,20]. However, this situation will not sustain as developing countries are gradually making policies to ensure their own environmental security. As it becomes more and more challenging to export plastic waste, developed countries will be constrained to reduce their consumption of plastic products at the source and improve their recycling technology and management of plastic waste[6,16,21]. As a ramification, the ban would extend from the reduction of waste trade flows to changes in the structure of treatment systems within different countries. Currently, the United States, the European Union, the United Kingdom, Australia, Japan, and Republic of Korea have proposed plans to reduce the consumption or increase recycling of plastic waste[20,22,23].

Apart from the Baseline and the aforementioned 2018 Scenario, we developed two types of prediction scenarios and a combination of the two scenarios to describe future trends. The "Exports Reduction Scenarios" assumes that the exports of plastic waste of developed countries and developing countries decrease tremendously, while the other, named "Recycling Rate Promotion Scenarios" presumes the recycling rates of countries increase gradually. The results of these are shown in Fig. 5.

Compared to the 2018 Scenario, reducing 50% of exports of all countries or only developed countries, or Zero exports Scenario all exacerbate the EIT for GW, but increased the beneficial environmental impact of three indicators (FPMF, FWE, and WC) to 128–197% besides raising the saved Eco-costs to about 1.35–1.73 folds (3.17–4.05 billion euros). The impacts of the China ban and other countries' subsequent policy measures[20] on the decline in global plastic waste trade flows are predictable, these changes will further improve the ecological environment compared to the Baseline and 2018 Scenario from a global environmental sustainability perspective. Although the Scenario cutting developing countries' exports by half turns around the positive EIT for GW, it's not the most efficient path. To reduce the environmental impacts for GW, the more effective solution emerges in the second kind of scenarios, as shown in Fig. 5. The environmental impacts for GW of 20%, 50%, and 100% increase recycling rate scenarios were 11%, −45%, and −34% of 2018 Scenario respectively, while the other four indicators show little changes. Therefore, not only the export reduction scenarios, but also the recycling rate promotion scenarios improved the current situation, mainly reflected by the capability of reversing the impact of GW. The combination scenarios, as optimization, balanced the impacts of these two scenarios. "Half of exports +20% increase recycling rate" scenario performs better than "Half of exports +50% increase recycling rate" scenario in FWE, HCT, WC, and Eco-cost, which indicates that the improvement in the recovery rate has its appropriate range. The main reason is that the manufacturing process of recycled plastic waste consumes more water than landfill and incineration. Therefore, "Half of exports +20% increase recycling rate" scenario is the most practical and effective pathway to mitigate environmental impacts of plastic waste trade flows worldwide in the future. Currently, the recycling rates of plastic waste are relatively low especially in developing countries (Supplementary Table 4). As all countries affected by the China ban gradually dispose of more plastic waste domestically and replace landfill or mismanagement with recycling, greater beneficial environmental impacts will be experienced on a long-term scale.

## Discussion

As per the analysis above, the China ban had remarkable impacts on the global plastic waste trade flow. In 2018, the trade flow of global plastic waste plummeted by 45.5%, and China's imports plunged by 95.4% compared to Baseline levels. The decreased amount of global trade flow and Chinese import flows had the same number of digits (6,503,977 tons for the world, and 7,596,188 tons for China). As the main alternative destinations, the total imports of Southeast Asia 5 surged to 362% against the Baseline Scenario. Nearly all major exporters were affected by the China ban, with total exports declining. We inferred that the higher the reliance of a country on the Chinese market (prior to the ban), the more dramatic will be the decrease in its plastic waste exports.

As a result of the China ban, in the short run, large amounts of waste will pile up or will be diverted to other developing countries like Southeast Asia 5, as most studies predicted[6,14,16]. Although the technologies of treating plastic waste in developed countries are more advanced, countries with high incineration rates have greater unit impact values for GW. As a result, the changes in trade flow had prompted negative but temporal environmental impacts on GW, but they had more or less improved the other four indicators (FPMF/ FWE/ HCT/ WC) and promoted global environmental sustainability. Overall, the trade flow changes saved a total of 2.35 billion euros of eco-cost annually around the globe following the China ban. It's worth mentioning that, with the improvement of import policies of developing countries, the export flows and recycling rates of plastic waste will further change[6,16,21]. It has been proved that reducing exports and increasing recycling rates are both effective measures to reduce the environmental footprints of plastic waste trades, scenarios of which save about 1.54–3.20 billion euros of eco-costs (excluding the ideal Zero exports Scenario). "Half of exports +20% increase recycling rate" scenario is the most recommended pathway taking into account the long-term environmental impacts.

Therefore, we suggest that developed countries should strengthen local management and treatment of waste through policy incentives and financial support rather than deliberately and recklessly exporting to foreign countries[24]. It is worth noting that the China ban only prohibits imports of plastic waste, but not recycled plastic components. If the domestic market capacity of developed countries is oversaturated, they can still export recycled plastic products to China, which is a big consumer of plastics. Developing countries need to raise awareness on the potential environmental risks of the disposal of foreign wastes and formulate related policies to thwart harmful repercussions. Most countries should further (1) restrict the production, sale, and use of some plastic products via regulations, (2) improve their classification of source of plastic waste by adopting educational, encouraging, and compulsory measures for residents and

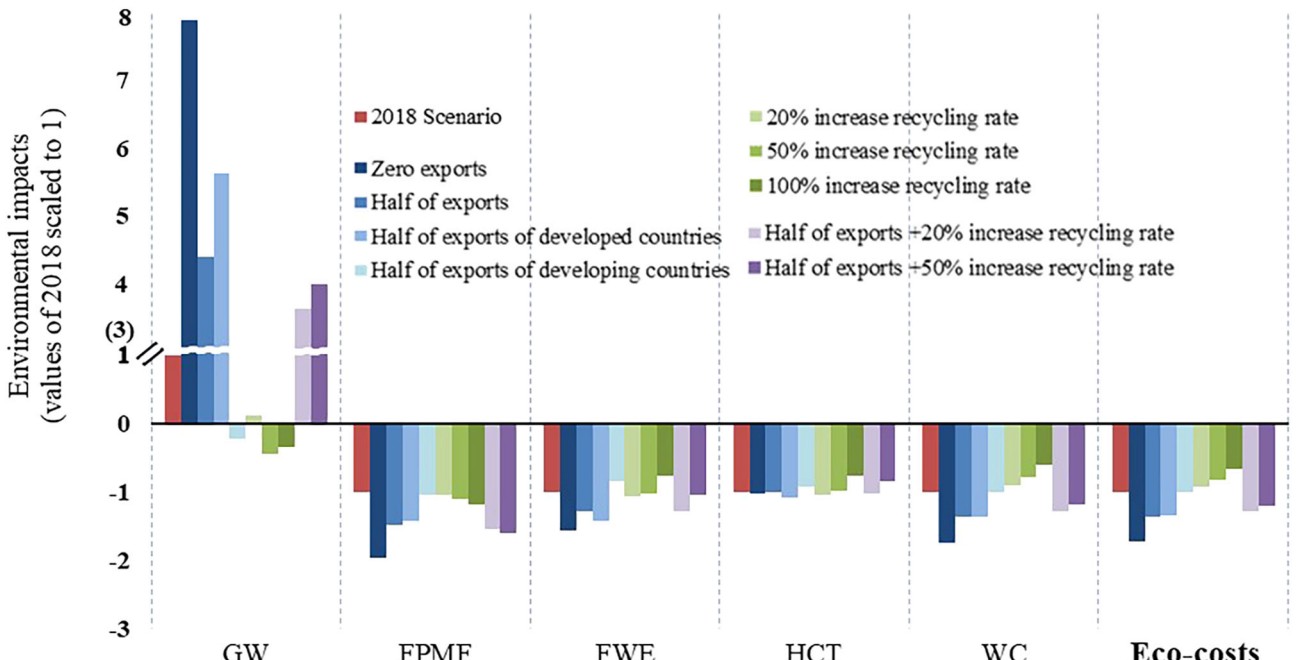

**Fig. 5 Comparison of the environmental impacts of Exports Reduction Scenarios, Recycling Rate Promotion Scenarios, and Combination Scenarios.** The data source can be found in Supplementary Table 9. The absolute value of each bar is equal to the absolute result of the actual value divided by its 2018 value.

enterprises, and (3) gradually raise the recycling rate of plastic waste[2,25−27] through the construction and improvement of recycling facilities. For example, China protocoled the "opinions on further strengthening the control of plastic pollution" in January 16, 2020, in a bid to actively tackle plastic pollution, and has ultimately been transitioning from a mixed-waste collection system to a source-segregation collection system[28].

## Methods

**Data and definition**. Most trade data were harnessed from The United Nations Commodity Trade Statistics Database (referred to as "Comtrade")[18], which includes detailed import and export trade data reported by governments of nearly 200 countries and regions. On the basis of the purposes of this study, we collected the plastic waste data of China, "17 countries" which accounted for more than 92% of total plastic waste imports to China (Hong Kong, Japan, the United States, seven countries in Europe (Germany, the United Kingdom, Belgium, Spain, Italy, France, and Netherlands, herein referred to as "Europe 7"), five countries in Southeast Asia (Thailand, Indonesia, Vietnam, Malaysia, and Philippines, herein referred to as "Southeast Asia 5"), Republic of Korea, and Mexico), as well as "other countries". Considering the bilateral trade asymmetries existing in international trade mirror data[29], the trade data are all based on exports to maintain consistency, unless the export data are unavailable.

Research statistics are divided into the data of Baseline Scenario and 2018 Scenario. In the Baseline Scenario, historical data of 9 years from 2008 to 2016 were used to describe the general scenario before the ban was issued. Although the ban came into effect at the beginning of 2018, it was announced in July 2017 and started to affect the waste trade market since then. The total import flows of China from 2008 to 2016 averaged $7.9 \times 10^6$ tons (from $6.9 \times 10^6$ to $8.9 \times 10^6$ tons), and dropped sharply to $5.4 \times 10^6$ tons in 2017, lower than any year in the Baseline Scenario. The imports of Hong Kong in 2017 were similar, which were only 48.6% of the 2008−2016 average and be the lowest in a decade. From 2008 to 2016, the import flows of Southeast Asia 5 basically showed a trend of year-on-year increase (from $2.6 \times 10^5$ to $8.1 \times 10^5$ tons), with a good linear fitting effect ($R = 0.936$) and an average annual growth rate of 15.2%. According to this trend, the imports of Southeast Asia 5 in 2017 should be around $9.3 \times 10^5$ tons. However, the actual import volume of Southeast Asia 5 in 2017 was as high as $1.5 \times 10^6$ tons, which was 1.87 times that of 2016. In addition, the USA, Japan, and Europe 7 began to reduce their plastic waste exports to China to varying degrees in 2017 and increase their exports to Southeast Asia. That's why the year 2017 was excluded in calculations under the Baseline scenario. As trade data fluctuates constantly, using the waste trade scale of a certain year to describe the Baseline Scenario will be inaccurate. In the situation where the data of the 9 years (from 2008 to 2016) fluctuate slightly and show an obvious trend, we used the average value of trade data from 2013 to 2016 to represent the Baseline Scenario; if it fluctuates greatly with no obvious

future trend projection, the data of all 9 years is considered, and the average calculated to represent the Baseline Scenario. As for the data of the 2018 Scenario, we mainly adopted the reporting data from Comtrade. Due to the delay in reporting the latest year data by individual countries, some missing data were estimated on the basis of monthly data, mirror data, and others.

$\lg(TF^{2018}/TF^{baseline})$ is a base-10 logarithm of the trade flows of the 2018 Scenario divided by the trade flows of Baseline Scenario of a certain country. The larger the positive value of $\lg(TF^{2018}/TF^{baseline})$, the greater the coefficient ratio of the 2018 Scenario to the Baseline. The trade flow change rate (TFCR) of every trade flow is the rate of changes in trade flows in the 2018 Scenario relative to the Baseline. To identify the relationship between the export change rate and dependence on Chinese market of a country, Export dependence on Chinese market (ED) is defined as the proportion of export flows to mainland China and Hong Kong in the total exports of a country, based on the baseline data (For Hong Kong, ED is defined as the proportion of exports to mainland China in its total exports). Calculations for $\lg(TF^{2018}/TF^{baseline})$, TFCR, and ED can be expressed mathematically as in Eq. (2)-(4):

$$\lg(TF^{2018}/TF^{baseline}) = \log_{10}\frac{TF_{2018}}{TF_{baseline}} \qquad (2)$$

$$TFCR = \frac{TF_{2018} - TF_{baseline}}{TF_{baseline}} \qquad (3)$$

$$ED = \left(\frac{TF_{export\ to\ China}}{TF_{export}}\right)_{baseline} \qquad (4)$$

As for the data for LCA, because of the differences among the environmental footprints of disposing various plastics, the plastic waste was divided into six types, namely polyethylene (PE), polystyrene (PS), polyvinyl chloride (PVC), polyethylene terephthalate (PET), polypropylene (PP) and others. The proportions of types of plastics and three common treatment options (including landfill (or mismanagement), incineration, and recycling) were based on various reports and research articles (Supplementary Table 4). The distance of ocean freighter transport was obtained from website [https://sea-distances.org]. Road transport distances for both importers and exporters and for domestic management were estimated to be 100 km[8]. The amount of electricity from incinerating 1 kg of plastic was set to 0.9 kWh for all types of plastic[30]. All the data of environmental impacts was obtained from SimaPro 8.5.2, an advanced LCA software developed by PRé sustainability (Supplementary Table 8). As suggested by documentation inside SimaPro, the electricity consumption for recycling 1 kg of any kind of plastic waste was set to 0.6 kWh globally. About 0.91 kg of recycled plastics, which were regarded as avoided products and would replace the same mass of virgin plastics, can be produced from 1 kg of plastic waste of the same kind based on recycling process reference data from Ecoinvent. This value has been applied in the previous study[8]. The electricity consumption and avoided products here are set uniformly

based on the suggestions of SimaPro and previous researches, which may vary with types and countries and lead to uncertainty within acceptance.

**Scenarios and method**. LCA as a method for quantifying and analyzing the potential environmental impacts over the life cycle of a product system[31], is recognized to be comprehensive, scientific, and well-developed[32], and has been applied in quantitative studies on plastic waste treatment[8,9,33,34]. To explore the environmental impacts of the changes of global plastic waste trade flow resulting from the China ban, we took into account resource and energy consumption indicators, toxic effects indicators, and common environmental damage indicators. As previous studies confirmed, the contribution of the Chinese plastic recycling industries to $CO_2$ emission reduction has been increasingly significant in recent years[35]. A large fraction of organics in condensable particulate matter emitted from waste incineration for power generation plants is derived from PVC in the waste[36]. The processes of plastic waste treatment impact freshwater ecosystems[37] and human health[38], and lead to changes in water intake[39]. Therefore, five of 18 midpoint indicators in ReCiPe which attract lots of attention were chosen for analysis, namely global warming (GW) (kg $CO_2$ eq), fine particulate matter formation (FPMF) (kg PM2.5 eq), freshwater ecotoxicity (FWE) (kg 1,4-DCB eq), human carcinogenic toxicity (HCT) (kg 1,4-DCB eq) and water consumption (WC) (m^3).

Eco-cost method, is a measure to estimate the average marginal prevention costs of midpoints of different impact categories, and has the advantage that the output is expressed in an intelligible monetary value (2017 Euro, €). The so-called prevention cost is the economic cost to reduce environmental pollution and materials depletion. For example, in order to meet the needs of human production and life while preventing 1000 kg of carbon dioxide emissions, it is necessary to invest 116 euros in offshore windmills. In other words, "the eco-costs of 1,000 kg carbon dioxide are € 116."

Calculations for Eco-cost can be expressed as in Eq. (5):

$$Eco-cost = \sum_i (EIT_i \times Eco_i) \qquad (5)$$

where i is the midpoint indicator in this study, $Eco$ represents the unit eco-cost value (and its upper and lower) of each midpoint (Supplementary Table 5), obtained from open databases[19]. The variation of unit eco-cost values leads to uncertainty.

Apart from the Baseline Scenario and the 2018 Scenario based on factual data, prediction scenarios are described as follows:

(1) Exports Reduction Scenarios: Half of exports represents the situation that exports of all countries decrease by 50% on basis of 2018 Scenario, while the scenarios named Half of exports of developed countries and Half of exports of developing countries represent situations that exports of some countries reduced by half and with the rest remaining unchanged. Zero exports indicate the scenario in which the export flows of all countries fall to 0 and all plastic waste is treated and disposed domestically. (2) Recycling Rate Promotion Scenarios: 20% increase recycling rate, 50% increase recycling rate and 100% increase recycling rate assumed that the recycling rates of all countries increased by 20%, 50% and 100% of 2018 Scenario with the same incineration rates (If the sum of incineration and recycling rates exceeds 100%, the recycling rate is equal to 100% minus incineration rate). (3) Combination Scenarios: Half of exports+20% (50%) increase recycling rate is the situation that exports of all countries decrease by 50% and the recycling rates of all countries increased by 20% (50%) on basis of the 2018 Scenario.

**Reporting summary**. Further information on research design is available in the Nature Research Reporting Summary linked to this article.

## Data availability

The datasets generated during and/or analyzed during the current study are included in this published article (and its supplementary information files) or be available from the corresponding author on reasonable request. The source data underlying Figs. 1, 2, 3, 4, and 5 and Supplementary Figs. 1, 2 3, 4, 5, and 6 are provided as a Source Data folder.

## Code availability

No codes were generated during the current study, and all software can be accessed to repeat the results of the current study.

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

## Acknowledgements

We would like to acknowledge the financial support from the National Science Foundation of China for Distinguished Young Scholars of China (71825006) and the "Thirteenth Five-Year" National Key Research and Development Program of China (2018YFC1903000). The contents of this paper reflect the views of the authors and do not necessarily indicate acceptance by the sponsors.

## Author contributions

Z.W. conceived the study, conducted data collection and technical analyses with the assistance of Y.X. and M.C. Y.X. drafted the paper, and revised the paper with input and feedback from Z.W. C.D.D. participated in the revision of the paper.

## Competing interests

The authors declare no competing interests.
