## [Peer Review File · Nature Communications]

REVIEWER COMMENTS

Reviewer #1 (Remarks to the Author):

The article is presenting the environmental impacts, in a short-term perspective, of the Chinese ban on plastic waste imports that was initiated in 2018. The researchers used the Life Cycle Assessment (LCA) method to quantify the environmental impacts of the trade flow changes and treatment methods of plastic waste (by type) in developed and developing countries (selected 18 countries) at the time point subsequent to the implementation of the ban in 2018. Drawing from the results of the LCA, the authors concluded that to achieve global environmental sustainability in a longer term perspective, countries should gradually move from export of plastic waste to domestic management, and from landfill to recycling. The quantitative results support the conclusions.

In general, the methodology used in this article is robust and adequate to explain the observed phenomena and reach potentially credible conclusions. However, there are several technical and theoretical discrepancies which must be addressed by the authors for increasing the credibility of the results and the reliability of the study. Firstly, I will point out the identified weaknesses in the different elements of the assessment methodology in the general comments for this article, and then I will conclude with some specific comments directly on the text of the manuscript (indicating line number). Finally, the article would benefit from a language proofreading, since several grammar and syntax mistakes pop up all over the manuscript.

General comments:

The article compares a baseline (2008-2016) with the situation in 2018. Firstly, why not including trade data for 2017? There is no explanation why 2017 data was excluded. On the contrary, a time period of 2008-2017 makes a round 10 year average value, that would fit very well as a baseline scenario. If there is a good reason to exclude 2017 data, I would encourage the authors to explain clearly in the manuscript.

Further, the assessment includes only data for 2018 in the "subsequent to the ban" scenario. Therefore, it presents only a snapshot of data (2018), but based on the results it makes further assumptions about the impact of the China ban which are not fully supported in reality. The 2018 scenario can indeed capture the actual situation, precisely after the ban, to illustrate the trade flow changes at that point in time. However, it does not account for the potential waste market restructuring delay which resulted in the redirection of plastic waste to SE Asian markets. Just after the ban, some plastic waste volumes did find its way to SE Asian markets, but a lot of plastic waste was stranded in countries of origin. There was a high stockpile of plastic waste for recycling that could not be redirected for recycling domestically and it was "temporarily stored" in landfills or other transfer facilities. Once the market normalised, due to the opening of more SE Asian capacity, the plastic waste trade resumed without China as a big player – although in lower volumes. Just looking at the UN comtrade data for 2019, it is obvious that the plastic waste trade was tripled (at least) with SE Asian countries, compared to 2018. Although the article does a good effort to capture the environmental impacts of the trade flow changes right after the ban was imposed, it does not reflect on the anticipated developments in 2019. To be fair, the article DOES discuss this potential outcome in the discussion part and offers pathways for strengthening the plastic ban approach throughout the region of SE Asia and other developing countries. However, the quantification of the environmental impacts of the China ban in this article could only measure the immediate (short-term) effects and could not account for the rebound of the policy. Would it be possible for the authors to take into consideration this concern and revise the modelling? Alternatively, I would suggest that the authors could run an additional scenario for 2019, where they can illustrate the changes of trade flows again, and how these have affected global environmental impacts. In this way, it would become evident that similar bans need to be imposed in other developing countries as well, and it will greatly strengthen the policy recommendations

presented in the discussion at the end of the article.

Concerning the technical characteristics of the assessment parameters, there are few issues to take into account:

1) In table b4, you present the share of the three common treatment options for plastic waste in each country. This data reflects the actual characteristics of waste management systems in each country prior to the implementation of the China ban. However, in order to calculate the environmental impacts of the trade flow change, you are estimating changes in the waste management options of the exporting countries, which are "forced" to treat plastic waste domestically. But you do not present your assumptions about the new % of waste treatment options per country. How these change after the ban? What are the precise numbers and how do you justify your assumptions?

2) As mentioned before, amounts of plastic waste collected for recycling would be stockpiled until a recycling solution is found. Large quantities of plastics could be stored in temporary facilities for up to a year and then shipped away for appropriate treatment. Alternatively, if no solution is found, the largest share of the plastic waste (separately collected for recycling) will not be redirected to municipal solid waste incineration, but it will be turned into RDF for use in industrial facilities. Thus substituting other fossil fuel sources, such as coal and oil. Therefore, the assessment for the GW indicator must be revised to account for this substitution effect, the same way as it is applied for the recycling (substituting virgin plastics).

3) In the treatment options, you include the category "landfilling and mismanagement". These two treatment options have significant differences, especially in terms of ecotoxicity and eutrophication. I am not sure if the data exist in the LCA databases to make the distinction between the two. If the data exist, then please separate the categories. If not, then please indicate clearly that these two management options have different environmental impacts but because of lack of data you decided to lump them up together. The LCA data you used in the assessment (table b8) indicate the process "sanitary landfill", which is distinctly different than disposal on land or water. Please revise the dataset if possible, otherwise indicate the limitations of your results.

4) You have chosen to run the LCA using 5 midpoint indicators. It would be a great improvement in the article, if you would clarify your reasoning behind the choice of these specific indicators.

5) Figure 1(a), in the diagram of the baseline you present HK as importer and exporter. The share of exports from HK is just a fraction of what appears in the graph, as it is apparent that HK is only re-exporting the imported waste from other countries and does not have domestic capacity for treatment. The graph is not wrong, but it shows the distorting effect of HK as a port for mainland China. For technical reasons, I believe there is no other way to present the graph, but you could again refer to the text about this effect and that plastic waste imported to HK have an (nearly) exclusive destination for China. Another suggestion would be to group them together and use the label "China and HK" or similar.

Specific comments:

1) In the abstract, please revise the first sentence. As it stands now, it is understood as there is only two major ways of dealing with plastic waste pollution – domestic management and exportation – since the late 1990s. While in reality, I think that the authors meant that since the late 1990s the trend of plastic waste shipment from developed to developing countries has been increasing. The sentence needs revision to reflect this meaning.

2) In the abstract, in the second sentence, you mention "plastic waste flow". It should be revised as "plastic waste trade flow". The prime objective of the article is trade flows and not domestic

flows. By adding "trade" in this phrase would increase clarity of the text.

3) In the abstract, last word of the second sentence, "undetermined" does not make sense in this context. Would it be better to replace with "unexamined" or "not researched", or similar?

4) In the introduction, line 2, what is a major material? Maybe use another word to describe the importance of plastic in the modern economy.

5) In line 12, you mention that plastic is a recyclable material. Not all plastics are recyclable. Better replace with a phrase that indicates that, e.g. most of plastic is recyclable.

6) In lines 32-33, maybe rephrase this sentence. The sentence does not come from the previous paragraph directly, so it cannot be claimed as a consequence using the transition word "therefore".

7) In line 34, you may replace the words "present researches" with "current research".

8) In line 69, the percentages presented are misleading. You mention "the recycling rates of all countries are raised to 120%, 150%, and 200%". This means that the actual recycling rate will become 120% or 200%. This is not correct, because a proportionate rate (e.g. recycling) can never be over 100% in absolute terms. What you really mean here is the increase of the recycling rate by 20%, 50% and 100% (essentially doubling the recycling rate) in comparison to the 2018 scenario. Therefore, you need to change this sentence (here and in the rest of the manuscript) so that it is understood correctly by the reader.

9) In lines 321-334, the discussion includes very good points and it is very important to highlight as your conclusions of the study. Well done!

10) In lines 332-354, the content of these paragraphs is not related to the findings of this study and to the topic of waste trade flows. The internal policies for managing plastic waste in China, other than the trade ban, are not within the scope of the research, as presented so far by the authors. As a result, these paragraphs seem to be completely unconnected to the rest of the manuscript and do not add anything significant to the conclusions of the research. These two paragraphs can be omitted.

11) In line 375, equation 3, the denominator seems to be wrong. Is it TF baseline instead of TV baseline ? If so, please correct it.

12) In line 411, the electricity consumption for recycling 1 kg of any kind of plastic waste was set to 0.6 kWh. Is this specific for the electricity mix of China? Please make clear what this value is representing.

13) In line 411, 0.91 kg of recycled plastics can be produced from 1kg waste plastics. This value seems quite optimistic. With the current technologies of mechanical recycling, the recycling yields can be lower (depending on the polymer type and purity of the waste stream). Please expand the review on background data, if possible.

14) In line 431, in the parenthesis, there is the word "lorhh". Please correct this word.

Reviewer #2 (Remarks to the Author):

This paper studied the environmental impacts of China's waste ban and discussed relevant policies. I find the analysis novel and interesting. However, I think the authors need to better describe and maybe improve their methodology and also improve the quality of the presentation before the

paper can be accepted. My specific comments are as follows.

First, a more explicit analysis of the production and trade patterns of waste plastics is needed. Overall, the waste ban policy could trigger two effects: First, changes in production levels of plastic waste in each country, and second, a change in the trade pattern among countries. Figure 1 (without any description of the scale) is unclear for understanding either aspect. Does it describe both interregional waste flows and regional waste productions, or only the former? Does the lower height of the "2018 Scenario" graph reflects less waste production due to China's policy, and if yes, what mechanism is responsible? Furthermore, how could the abovementioned two types of change contribute to the environmental impacts of the waste ban? The results are convincing only if there is a reasonable quantification of changes in the production and trade patterns.

Second, a better description of the differences in waste treatment technologies between countries is needed. I think those are the main effects driving the evaluation of environmental impacts due to the trade ban. Although the authors used standard LCA techniques, these methods are subject to considerable uncertainties (which are also not presented). What technologies in what countries contribute most to the overall decrease in environmental impacts in Figure 4? And has the waste ban triggered any change in relevant technology (not only because of the rising cost of disposing of wastes, but also the change in the scale of waste each country has to deal with)? The paper needs to focus on these substantial issues besides describing the numbers from LCA evaluations.

Third, the paper needs improvement in writing, especially concerning punctuations. Examples can be found in line 35, 42, 45, 61 and 123. Also, to match the overall paper quality of this journal, I think the author should make Figure 2, 4 and 5 simply look better. In line 171, it should not be the R2 value being -0.677.

Overall, the paper needs to capture better the complex interactions between policy, production, trade, and technology, and improve the presentation accordingly. Only then could it provide a convincing analysis of this very interesting topic.

Dear Editors and Reviewers:

Thanks for your comments! Your insightful comments and suggestions are highly appreciated. The issues raised by the reviewers have been addressed accordingly and responses to each comment are given below. We hope that the revision has responded to the reviewer's comments and is now acceptable for publication in your journal.

Sincerely,
Zongguo Wen, on behalf of the authors

RESPONSES TO REFEREES

Comment	Response
Reviewer 1	
Why not including trade data for 2017? There is no explanation why 2017 data was excluded. On the contrary, a time period of 2008-2017 makes a round 10 year average value, that would fit very well as a baseline scenario. If there is a good reason to exclude 2017 data, I would encourage the authors to explain clearly in the manuscript.	The reason for excluding 2017 is that the ban is announced in July 2017 and started to affect the global market since then. We have added corresponding explanation in lines 391-394.
Further, the assessment includes only data for 2018 in the "subsequent to the ban" scenario. Therefore, it presents only a snapshot of data (2018), but based on the results it makes further assumptions about the impact of the China ban which are not fully supported in reality. The 2018 scenario can indeed capture the actual situation, precisely after the ban, to illustrate the trade flow changes at that point in time. However, it does not account for the potential waste market restructuring delay which resulted in the redirection of plastic waste to SE Asian markets. Just after the ban, some plastic waste volumes did find its way to SE Asian markets, but a lot of plastic waste was stranded in countries of origin. There was a high stockpile of plastic waste for recycling that could not be redirected for recycling domestically and it was "temporarily stored" in landfills or other transfer facilities.	It is meaningful to analyse the trade data of 2019 so as to consider the potential waste market restructuring delay. However, there have been only 8 of 18 sample countries which reported their export data of 2019 up to May 26, 2020 as shown in the New Tab.b10. We tried to make up for the missing part with mirrored data and monthly data, but it is still not complete and brings in uncertainties that cannot be ignored. Therefore, we failed to establish the 2019 scenario based on the data available, and decided to add a qualitative discussion in the manuscript (lines 141-151). Because None of the Southeast Asia 5 reported import data for 2019, we calculated the plastic waste trade flows from all countries that have export data for both 2018 and 2019 to Southeast Asia 5, and found that the trade flows in 2019 to Southeast Asia 5 decreased by 32% compared to 2018. This result is consistent with the 8 sample countries that have reported export data to Southeast Asia 5 of 2019 (69% of 2018), as shown in

Once the market normalised, due to the opening of more SE Asian capacity, the plastic waste trade resumed without China as a big player – although in lower volumes. Just looking at the UN comtrade data for 2019, it is obvious that the plastic waste trade was tripled (at least) with SE Asian countries, compared to 2018. Although the article does a good effort to capture the environmental impacts of the trade flow changes right after the ban was imposed, it does not reflect on the anticipated developments in 2019. To be fair, the article DOES discuss this potential outcome in the discussion part and offers pathways for strengthening the plastic ban approach throughout the region of SE Asia and other developing countries. However, the quantification of the environmental impacts of the China ban in this article could only measure the immediate (short-term) effects and could not account for the rebound of the policy. Would it be possible for the authors to take into consideration this concern and revise the modelling? Alternatively, I would suggest that the authors could run an additional scenario for 2019, where they can illustrate the changes of trade flows again, and how these have affected global environmental impacts. In this way, it would become evident that similar bans need to be imposed in other developing countries as well, and it will greatly strengthen the policy recommendations presented in the discussion at the end of the article.

the following table. We're afraid that we may misunderstand the sentence in your comment: "Just looking at the UN comtrade data for 2019, it is obvious that the plastic waste trade was tripled (at least) with SE Asian countries, compared to 2018". The import flows of Southeast Asia 5 in 2018 exactly rose to 3.62 times that of the Baseline Scenario, however this trend didn't continue in 2019, because southeast Asian countries have gradually issued import restriction policies. We'd like to deepen our work when this misunderstanding is clarified.

Exporter	Importer	A1:trade volume in 2019 (t)	A2:trade volume in 2018 (t)	A1/A2
HK, China	southeast Asia 5	226185	225430	1.00335
Japan	southeast Asia 5	511530	563416	0.90791
USA	southeast Asia 5	141691	422323	0.3355
Germany	southeast Asia 5	233606	255173	0.91548
UK	southeast Asia 5	23885	213167	0.11205
Belgium	southeast Asia 5	93755	74937	1.25112
Italy	southeast Asia 5	12469	22275	0.55978
Rep. of Korea	southeast Asia 5	26738	53461	0.50014
8 sample countries	southeast Asia 5	1269859	1830182	0.69384
all reporter	southeast Asia 5	1331851	1948554	0.68351

In table b4, you present the share of the three common treatment options for plastic waste in each country. This data reflects the actual characteristics of waste management systems in each country prior to the implementation of the China ban. However, in order to calculate the environmental impacts of the trade flow change, you are estimating changes in the waste management options of the exporting countries, which are "forced" to treat plastic waste domestically. But you do not present your assumptions about the new % of waste treatment

We conducted additional research on the data of new % of waste treatment options per countries. There are 10 of 18 sample countries (Europe 7, China, Japan and USA) having published their new % of 2018 in official reports. It is showed that the landfill rate of these countries declined, recycling rate rose and incineration rate fluctuated between baseline to 2018 (new Tab.b4). In general, the magnitude of the changes in the ratio was slight, so for the other 8 countries (and regions) that the latest % are not available, we assumed that the shares remained unchanged and recalculate the results. The corresponding improvements are shown in Tab.b4,

options per country. How these change after the ban? What are the precise numbers and how do you justify your assumptions?	lines 199-206 and the data results after lines 174.
As mentioned before, amounts of plastic waste collected for recycling would be stockpiled until a recycling solution is found. Large quantities of plastics could be stored in temporary facilities for up to a year and then shipped away for appropriate treatment. Alternatively, if no solution is found, the largest share of the plastic waste (separately collected for recycling) will not be redirected to municipal solid waste incineration, but it will be turned into RDF for use in industrial facilities. Thus substituting other fossil fuel sources, such as coal and oil. Therefore, the assessment for the GW indicator must be revised to account for this substitution effect, the same way as it is applied for the recycling (substituting virgin plastics).	It is a potential pathway for the production of RDF from plastic waste to avoiding the need to use coal or oil as auxiliary fuel in waste incineration plants. Electricity production from RDF incineration is another form of energy recovery, the environmental impacts due to GW of which are similar to direct waste incineration, as suggested by previous studies^{1,2}. The application rate of this technical route is not high, for instance, there were 13 RDF incinerators in the USA, whose treatment capacity accounted for 3.64% of domestic waste in 2014³. There is also more environmentally friendly application of RDF (e.g. the utilization of RDF in cement production and biofuels production), but their shares in practice are currently small^{1,3,4}. For example, the domestic waste used as RDF for cement production accounts for 1.4% of China's total domestic waste in 2015, 0.6% in Germany in 2014 and 0.22% in Switzerland in 2014. There is no denying that RDF made from plastic waste and other household waste has great popularization potential in the future. However, for the difficulty of further classifying the small share of these advanced pathways of all sample countries, we decided to add a discussion of these possible plastic waste treatment pathways to the manuscript to make an improvement, which is reflected in lines 188-190.
In the treatment options, you include the category "landfilling and mismanagement". These two treatment options have significant differences, especially in terms of ecotoxicity and eutrophication. I am not sure if the data exist in the LCA databases to make the distinction between the two. If the data exist, then please separate the categories. If not, then please indicate clearly that these two management options have different environmental impacts but because of lack of data you decided to lump them up together. The LCA data you used in the	Thanks for pointing out the limitation here. The most common form of plastic waste mismanagement is open dumping, mainly visible in low-income countries, which may lead to more serious environmental impacts than sanitary landfill. However, exactly as mentioned, we failed to obtain the end-of-life environmental impacts of mismanagement in LCA databases and the share of mismanagement has not been recorded in official reports. Therefore, we decided to take landfills to approximate mismanagement. We have added explanation in the manuscript to indicate the limitation, which is reflected in lines 186-188.

assessment (table b8) indicate the process “sanitary landfill”, which is distinctly different than disposal on land or water. Please revise the dataset if possible, otherwise indicate the limitations of your results.	
You have chosen to run the LCA using 5 midpoint indicators. It would be a great improvement in the article, if you would clarify your reasoning behind the choice of these specific indicators.	Thanks for your suggestion. As per the added explanation in lines 446-456, we took into account resource and energy consumption indicators, toxic effects indicators and common environmental damage indicators. The processes of plastic waste treatments impact on CO₂ emission⁵, fine particulate matter formation⁶, freshwater ecosystems⁷, human health⁸, and water intake⁹, as previous studies confirmed. Therefore, five of 18 midpoint indicators in ReCiPe which attract lots of attention were chosen for analysis.
Figure 1(a), in the diagram of the baseline you present HK as importer and exporter. The share of exports from HK is just a fraction of what appears in the graph, as it is apparent that HK is only re-exporting the imported waste from other countries and does not have domestic capacity for treatment. The graph is not wrong, but it shows the distorting effect of HK as a port for mainland China. For technical reasons, I believe there is no other way to present the graph, but you could again refer to the text about this effect and that plastic waste imported to HK have an (nearly) exclusive destination for China. Another suggestion would be to group them together and use the label “China and HK” or similar.	Thanks for highlighting the distorting effect of HK as a port for mainland China. We have grouped “China mainland” and “HK” together as “China mainland and HK” in new Figure 1.
In the abstract, please revise the first sentence. As it stands now, it is understood as there is only two major ways of dealing with plastic waste pollution – domestic management and exportation – since the late 1990s. While in reality, I think that the authors meant that since the late 1990s the trend of plastic waste shipment from developed to developing countries has been increasing. The sentence needs revision to reflect this meaning.	Thank you! This has been modified according to your suggestion.

In the abstract, in the second sentence, you mention “plastic waste flow” . It should be revised as “plastic waste trade flow” . The prime objective of the article is trade flows and not domestic flows. By adding “trade” in this phrase would increase clarity of the text.	Thank you! To maintain consistency, this has been modified in the abstract, text and title.
In the abstract, last word of the second sentence, “undetermined” does not make sense in this context. Would it be better to replace with “unexamined” or “not researched” , or similar?	Thank you! This has been modified.
In the introduction, line 2, what is a major material? Maybe use another word to describe the importance of plastic in the modern economy.	Thank you! This has been modified in line 2 with the phrase “important material” .
In line 12, you mention that plastic is a recyclable material . Not all plastics are recyclable. Better replace with a phrase that indicates that, e.g. most of plastic is recyclable.	Thank you! This has been modified in line 12.
In lines 32-33, maybe rephrase this sentence. The sentence does not come from the previous paragraph directly, so it cannot be claimed as a consequence using the transition word “therefore” .	Thank you! This has been modified in line 32 with the phrase “from a global sustainability perspective” .
In line 34, you may replace the words “present researches” with “current research” .	Thank you! This has been modified in line 35.
In line 69, the percentages presented are misleading. You mention “the recycling rates of all countries are raised to 120%, 150%, and 200%” . This means that the actual recycling rate will become 120% or 200%. This is not correct, because a proportionate rate (e.g. recycling) can never be over 100% in absolute terms. What you really mean here is the increase of the recycling rate by 20%, 50% and 100% (essentially doubling the recycling rate) in comparison to the 2018 scenario. Therefore, you need to change	Thank you! Our description here was misleading, and what we really mean is exactly as your explanation. This has been corrected in the manuscript in lines 70-71 and the rest of the manuscript.

this sentence (here and in the rest of the manuscript) so that it is understood correctly by the reader.	
In lines 332-354, the content of these paragraphs is not related to the findings of this study and to the topic of waste trade flows. The internal policies for managing plastic waste in China, other than the trade ban, are not within the scope of the research, as presented so far by the authors. As a result, these paragraphs seem to be completely unconnected to the rest of the manuscript and do not add anything significant to the conclusions of the research. These two paragraphs can be omitted.	Thank you! This has been deleted.
In line 375, equation 3, the denominator seems to be wrong. Is it TF baseline instead of TV baseline ? If so, please correct it.	Thank you! This has been corrected.
In line 411, the electricity consumption for recycling 1 kg of any kind of plastic waste was set to 0.6 kWh. Is this specific for the electricity mix of China? Please make clear what this value is representing.	It is true that the electricity consumption of plastic recycling varies with types and countries^{10,11}. It is assumed that 0.6 kWh of electricity is consumed to recycle and remanufacture 1 kg of plastic from plastic waste globally, as suggested by documentation in SimaPro. However, it is also emphasized in the documentation that this is only a rough estimate and this parameter is of great uncertainty. That's why we did sensitivity analysis in Tab. b7. For now, it is still hard to distinguish this value for different types of plastic waste in different countries. Therefore, we add corresponding limitations in lines 437-439.
In line 411, 0.91 kg of recycled plastics can be produced from 1kg waste plastics. This value seems quite optimistic. With the current technologies of mechanical recycling, the recycling yields can be lower (depending on the polymer type and purity of the waste stream). Please expand the review on background data, if possible.	This value is based on recycling process reference data from Ecoinvent and being applied in the previous study¹⁰. After receiving your suggestion, we expanded the review on background data and found that this value in different studies varies. A study put it above 0.95 kg¹², and another one adopted the data of 2005 and set this value to 0.88 kg for PE¹³. There are also literatures with lower values, for example 0.8 kg¹¹ and 0.73~0.88 kg in the 2000s¹⁴. We further consulted the

	China Scrap Plastics Association, and got that this value may vary from 0.7 to 0.95 kg due to different technologies and products. According to our results of sensitivity analysis, the sensitivity coefficient of this value is quite low for FPMF, FEW, HCT and WC. Therefore, we think that it is reasonable to set this value to 0.91 kg, and the error is within acceptable limits. We added the limitation in lines 437-439.
In line 431, in the parenthesis, there is the word “lorhh”. Please correct this word.	This has been corrected.
Reviewer 2	
First, a more explicit analysis of the production and trade patterns of waste plastics is needed. Overall, the waste ban policy could trigger two effects: First, changes in production levels of plastic waste in each country, and second, a change in the trade pattern among countries. Figure 1 (without any description of the scale) is unclear for understanding either aspect. Does it describe both interregional waste flows and regional waste productions, or only the former? Does the lower height of the “2018 Scenario” graph reflects less waste production due to China’s policy, and if yes, what mechanism is responsible? Furthermore, how could the abovementioned two types of change contribute to the environmental impacts of the waste ban? The results are convincing only if there is a reasonable quantification of changes in the production and trade patterns.	We hold the same opinion that the ban does lead to changes in generation levels of plastic wastes, for example, China protocolled the "opinions on further strengthening the control of plastic pollution" in 2020 to restrict the generation of plastic wastes from the production of plastic products on the list published in April 2020. But quantitative data on this change is lacking for now, and estimations from various sources are highly uncertain. Therefore, we chose to focus on the changes of waste trade flows in this research, which represent part of the changes in waste generation indirectly. Thanks to your suggestion, we added the scale of Figure 1, which describes interregional waste flows. The scales of Figure 1(a) and 1(b) are the same. Therefore, the lower height of the “2018 Scenario” graph reflects less waste trade flow due to the China ban, corresponding to the analysis in line 96. (The world’s total plastic waste trade flow plunged by 45.5% in 2018 compared to the Baseline Scenario.)
Second, a better description of the differences in waste treatment technologies between countries is needed. I think those are the main effects driving the evaluation of environmental impacts due to the trade ban. Although the authors used standard LCA techniques, these methods are	(1) The differences in waste treatment technologies are exactly the main sources of environment impacts of trade flow changes. We add a qualitative discussion about Tab. b4 of the differences between the treatment structures of developed and developing countries and changes before and after the ban in lines 199-206.

subject to considerable uncertainties (which are also not presented). What technologies in what countries contribute most to the overall decrease in environmental impacts in Figure 4? And has the waste ban triggered any change in relevant technology (not only because of the rising cost of disposing of wastes, but also the change in the scale of waste each country has to deal with)? The paper needs to focus on these substantial issues besides describing the numbers from LCA evaluations.	(2) The uncertainties of LCA is elaborated in lines 186-190 and 437-439. (3) To trace the contribution of countries on the overall decrease in environmental impacts in Figure 4, we added Fig. b5(b) and corresponding discussion of results in lines 240-256. (4) The ban had led to the changes of treatment structure of sample countries. We conducted additional research on the data of new rate of waste treatment options per countries as shown in the new Tab.b4. There are 10 of 18 sample countries (Europe 7, China, Japan and USA) having published their new rate of landfill, incineration and recycling of 2018 in official reports. In general, the magnitude of the changes in the ratio was slight, so for the other 8 countries (and regions) that the latest rate are not available, we assumed that the shares remained unchanged, and recalculate the results.
Third, the paper needs improvement in writing, especially concerning punctuations. Examples can be found in line 35, 42, 45, 61 and 123. Also, to match the overall paper quality of this journal, I think the author should make Figure 2, 4 and 5 simply look better. In line 171, it should not be the R2 value being -0.677.	(1) We have corrected the examples and invited a native English-speaking doctor to improve paper writing. (2) Figure 2, 4 and 5 have been improved. (3) The R value in original Line 171 represents the correlation between the export flow changes of countries and their dependence on the Chinese market. The R value is equal to -0.677, and the R² value is 0.4586. To avoid misunderstandings, we unify the values in the text and Figure 2 as R.

Reference

- 1 Liikanen, M., Havukainen, J., Viana, E. & Horttanainen, M. Steps towards more environmentally sustainable municipal solid waste management - A life cycle assessment study of Sao Paulo, Brazil. *Journal of Cleaner Production* 196, 150-162, doi:10.1016/j.jclepro.2018.06.005 (2018).
- 2 Havukainen, J. et al. Environmental impact assessment of municipal solid waste management incorporating mechanical treatment of waste and incineration in Hangzhou, China. *Journal of Cleaner Production* 141, 453-461, doi:10.1016/j.jclepro.2016.09.146 (2017).
- 3 Zhang, L. et al. Application of Cement Kiln Co-processing Municipal Solid Waste. *Environmental Sanitation Engineering* 25, 28-30 (2017).

- 4 Aracil, C., Haro, P., Giuntoli, J. & Ollero, P. Proving the climate benefit in the production of biofuels from municipal solid waste refuse in Europe. *Journal of Cleaner Production* 142, 2887-2900, doi:10.1016/j.jclepro.2016.10.181 (2017).
- 5 Liu, Z. et al. How does circular economy respond to greenhouse gas emissions reduction: An analysis of Chinese plastic recycling industries. *Renewable and Sustainable Energy Reviews* 91, 1162-1169, doi:https://doi.org/10.1016/j.rser.2018.04.038 (2018).
- 6 Wang, G., Deng, J., Ma, Z., Hao, J. & Jiang, J. Characteristics of filterable and condensable particulate matter emitted from two waste incineration power plants in China. *Science of The Total Environment* 639, 695-704, doi:https://doi.org/10.1016/j.scitotenv.2018.05.105 (2018).
- 7 Magni, S. et al. Plastics and biodegradable plastics: ecotoxicity comparison between polyvinylchloride and Mater-Bi® micro-debris in a freshwater biological model. *Science of The Total Environment* 720, 137602, doi:https://doi.org/10.1016/j.scitotenv.2020.137602 (2020).
- 8 Cook, C. R. & Halden, R. U. in *Plastic Waste and Recycling* (ed Trevor M. Letcher) 513-527 (Academic Press, 2020).
- 9 Hou, P. et al. Life cycle assessment of end-of-life treatments for plastic film waste. *Journal of Cleaner Production* 201, 1052-1060, doi:https://doi.org/10.1016/j.jclepro.2018.07.278 (2018).
- 10 Chen, Y. et al. Life cycle assessment of end-of-life treatments of waste plastics in China. *Resources Conservation and Recycling* 146, 348-357, doi:10.1016/j.resconrec.2019.03.011 (2019).
- 11 Horodytska, O., Kiritsis, D. & Fullana, A. Upcycling of printed plastic films: LCA analysis and effects on the circular economy. *Journal of Cleaner Production*, 122138, doi:https://doi.org/10.1016/j.jclepro.2020.122138 (2020).
- 12 Gu, F., Guo, J., Zhang, W., Summers, P. A. & Hall, P. From waste plastics to industrial raw materials: A life cycle assessment of mechanical plastic recycling practice based on a real-world case study. *Science of the Total Environment* 601, 1192-1207, doi:10.1016/j.scitotenv.2017.05.278 (2017).
- 13 Khoo, H. H. LCA of plastic waste recovery into recycled materials, energy and fuels in Singapore. *Resources Conservation and Recycling* 145, 67-77, doi:10.1016/j.resconrec.2019.02.010 (2019).
- 14 Nakatani, J., Fujii, M., Moriguchi, Y. & Hirao, M. Life-Cycle Assessment of Domestic and Transboundary Recycling of Post-Consumer PET Bottles. *Journal of Life Cycle Assessment, Japan* 4, 324-333, doi:10.3370/lca.4.324 (2008).

REVIEWER COMMENTS

Reviewer #1 (Remarks to the Author):

Dear authors,

Thank you for considering my previous comments on the manuscript and for the work you have done to improve the manuscript and to address satisfactorily (with sufficient clarifications) all the underlying assumptions in your calculations. However, there is still one issue that needs to be resolved before I could recommend the article for publication.

The argumentation why data from 2017 have been omitted from your baseline is not strongly supported. The justification is still very weak, since the trade ban is not in effect YET in 2017. Practically, there might be an early reorganisation of the market and reluctance from traders to engage in trade, but technically there is no constrain for trade operations, business as usual. Do you have any evidence, other than trade data, that would justify the distortion in the market as early as July 2017? How would you justify your conclusion that "the ban announced in July 2017 [...]started to affect the global market since then." ? Is there any other supportive evidence that can prove this is the case?

In the manuscript, you mention that for calculating the Baseline Scenario you firstly identified a trend between 2008-2016 and then calculated the average value of trade data from 2013 to 2016. With the inclusion of the 2017 data the general trend of the period 2008-2017 might change and therefore affect the calculation of the Baseline Scenario and thus the credibility of the total results of the five indicators.

For this reason, it is important to be able to justify strongly (and with supportive evidence) your decision to exclude data from 2017, so as to clear any doubt and uncertainty about the validity of the results of your study.

Reviewer #2 (Remarks to the Author):

Thanks for the authors for addressing my major concerns in the revision. The paper has been significantly improved. There are still some relatively minor problems to be solved before publication in this renowned journal.

The paper states that the trade flow changes could save specific amounts of money, in lines 16, 20, and 372. I understand this should come from some LCA software, but the paper should briefly discuss how these numbers are estimated, and what could be the limitations and uncertainties. And in what year's currency?

Please try not start a sentence with numbers (e.g., line 16).

Line 42: The Basel Convention has not been effective due to what?

Line 89: "in the future" is redundant after the word "long-term".

Line 266, 276, 302, 304, 361, 396: Please check grammar.

Line 381: "strongly ... strengthen" seems repetitive.

Dear Editors and Reviewers:

Thanks for your comments. Your insightful comments and suggestions are highly appreciated. The issues raised by the reviewers have been addressed accordingly and responses to each comment are given below. We hope the revision has responded to the reviewer's comments and is now acceptable for publication in your journal.

Sincerely,
Zongguo Wen, on behalf of the authors

RESPONSES TO REFEREES

Comment	Response
Reviewer 1	
The argumentation why data from 2017 have been omitted from your baseline is not strongly supported. The justification is still very weak, since the trade ban is not in effect YET in 2017. Practically, there might be an early reorganisation of the market and reluctance from traders to engage in trade, but technically there is no constrain for trade operations, business as usual. Do you have any evidence, other than trade data, that would justify the distortion in the market as early as July 2017? How would you justify your conclusion that "the ban announced in July 2017 [...] started to affect the global market since then." ? Is there any other supportive evidence that can prove this is the case? In the manuscript, you mention that for calculating the Baseline Scenario you firstly identified a trend between 2008-2016 and then calculated the average value of trade data from 2013 to 2016. With the inclusion of the 2017 data the general trend of the period 2008-2017 might change and therefore affect the calculation of the Baseline Scenario and thus the credibility of the total results of the five indicators. For this reason, it is important to be able to justify	Thank you for pointing out the shortcoming. To provide the evidence of excluding data from 2017, we added a discussion of trade flows of global plastic waste market in 2017 in lines 415-430, and put the trade data of 2017 in the new Tab. b11 in the Supplementary Information. As stated in the discussion, although the ban came into effect at the beginning of 2018, it was announced in July 2017 and started to affect the waste trade market since then. The total import flows of China from 2008 to 2016 averaged 7.9×10^6 tons each year (from 6.9×10^6 to 8.9×10^6 tons), and sharply dropped to 5.4×10^6 tons in 2017, lower than any year in the Baseline Scenario. Similarly, the imports of Hong Kong in 2017 were only 48.6% of the 2008-2016 average quote, which made it the lowest in that decade. From 2008 to 2016, the import flows of Southeast Asia 5 basically showed a trend of year-on-year increase (from 2.6×10^5 to 8.1×10^5 tons), with a good linear fitting effect ($R=0.936$) and an average annual growth rate of 15.2%. According to this trend, the imports of Southeast Asia 5 in 2017 should be around 9.3×10^5 tons. However, the actual import volume of Southeast Asia 5 in 2017 was as high as 1.5×10^6 tons, which was 1.87 times that of 2016. In addition, the USA, Japan and Europe 7 began to reduce their plastic waste exports to China to varying

strongly (and with supportive evidence) your decision to exclude data from 2017, so as to clear any doubt and uncertainty about the validity of the results of your study.	degrees in 2017 and increase their exports to Southeast Asia. The data analysis shows that the ban has had a evident impact on the import and export markets of major countries in 2017, so 2017 should be excluded from the baseline scenario.
Reviewer 2	
The paper states that the trade flow changes could save specific amounts of money, in lines 16, 20, and 372. I understand this should come from some LCA software, but the paper should briefly discuss how these numbers are estimated, and what could be the limitations and uncertainties. And in what year's currency?	Thanks for your suggestions. We have added a statement that all economic parameters were normalized to 2017 Euro (€) in line 497 and further explained the source of prevention cost numbers in lines 497-502. There is indeed uncertainty in the application of parameters. We used the upper and lower values of Eco-costs per kilogram of midpoints to calculate the upper and lower limits of total eco-costs respectively in tab.b5. The results showed that the variation of unit values brings certain uncertainty of eco-costs. It is one of the limitations of this paper. However, the eco-costs here are to provide a link between environmental benefits and economic benefits of the ban for intuitive understanding, so the uncertainty is acceptable. We further stated the limitations in line 507.
Please try not start a sentence with numbers (e.g., line 16).	Thank you. We have revised three sentences starting with numbers in the text, which are in the line 16, line 109 and line 468 respectively.
Line 42: The Basel Convention has not been effective due to what?	Firstly, the Basel Convention didn't completed ban the global trade of hazardous waste, but require prior informed consent for transboundary movements of hazardous wastes from potential importing and transit countries. Meanwhile, the convention does not offer a clear definition of hazardous waste, which makes it impossible to abide by. What's more, some major waste exporters such as the U.S. refuses to sign on to the Basel Convention. Therefore, some widely acknowledged hazardous waste is still being shipped from developed countries to developing countries, not to mention other types of waste such as plastic waste. An investigation directed by the National Trans-Frontier Shipment Office (NTSFO) in 2017 traced some electronic waste produced in Limerick, Ireland and finally located it in Hong Kong.

	For these reasons and evidences, we hold the opinion that the Basel Convention has not been effective. We have add the reasons briefly in lines 43-45.
Line 89: "in the future" is redundant after the word "long-term".	Thank you. We have removed the phrase "in the future".
Line 266, 276, 302, 304, 361, 396: Please check grammar.	Thank you. We have revised these sentences in line 266, lines 274-279, line 304, line 306, lines 363-365 and lines 395-398 respectively.
Line 381: "strongly ... strengthen" seems repetitive.	Thank you. We have removed the word "strongly".

REVIEWERS' COMMENTS

Reviewer #1 (Remarks to the Author):

Thank you for addressing adequately all data issues in the manuscript. It looks like a solid work now and I would gladly recommend for publication.

Reviewer #2 (Remarks to the Author):

The authors have carefully addressed my concerns.

Also, they explained why they have excluded the data on 2017 from the analysis, according to Reviewer #1's comments. Since the policy was announced in 2017, it already impacted the market, as can be seen from the newly added data. I understand the authors excluded these data in the main analysis to have a clear assessment of the policy's full impact (compared to the baseline). Therefore, I think the explanation is convincing.

Dear Reviewers:

Thanks for your approval and recommend of our paper.

Sincerely,

Zongguo Wen, on behalf of the authors

RESPONSES TO REFEREES

Comment	Response
Reviewer 1	
Thank you for addressing adequately all data issues in the manuscript. It looks like a solid work now and I would gladly recommend for publication.	Thank you for your previous suggestions on the 2017 data, which allowed us to improve the article to a large extent.
Reviewer 2	
The authors have carefully addressed my concerns. Also, they explained why they have excluded the data on 2017 from the analysis, according to Reviewer #1's comments. Since the policy was announced in 2017, it already impacted the market, as can be seen from the newly added data. I understand the authors excluded these data in the main analysis to have a clear assessment of the policy's full impact (compared to the baseline). Therefore, I think the explanation is convincing.	Thank you for your previous detailed suggestions on eco-cost, the Basel Convention and writing specifications. At the same time, thank you for your approval of our further data interpretation for the 2017 data.